# PROCEEDINGS A

complexity, biophysics, chaos theory

Jacobian matrix, early warning signals, meta-community, foodweb

**Author for correspondence:**
Thilo Gross
e-mail: thilo2gross@gmail.com

†These authors contributed equally to this study.

# A closed form for Jacobian reconstruction from time series and its application as an early warning signal in network dynamics

Edmund Barter[1,†], Andreas Brechtel[2,†],
Barbara Drossel[2] and Thilo Gross[3,4,5,6]

[1]Department of Engineering Mathematics, University of Bristol, Merchant Venturers Building, Woodland Road, Bristol BS8 1UB, UK
[2]TU Darmstadt, Fachbereich Physik, Hochschulstrasse, 6, 64289 Darmstadt, Germany
[3]HIFMB Helmholtz Institute for Functional Marine Biodiversity, Ammerländer Heerstr. 231, Oldenburg, Germany
[4]Alfred-Wegener-Institute for Marine and Polar Research, Am Handelshaven 12, Bremerhaven, Germany
[5]University of Oldenburg, Institute for Chemistry and Biology of the Marine Environment, Carl-von-Ossietzky 9-11 Str., Germany
[6]UC Davis, Department of Computer Science, 1 Shields Avenue Davis, CA 95616, USA

(iD) EB, 0000-0002-0165-0633; TG, 0000-0002-1356-6690

The Jacobian matrix of a dynamical system describes its response to perturbations. Conversely, one can estimate the Jacobian matrix by carefully monitoring how the system responds to environmental noise. We present a closed-form analytical solution for the calculation of a system's Jacobian from a time series. Being able to access the Jacobian enables a broad range of mathematical analyses by which deeper insights into the system can be gained. Here we consider in particular the computation of the leading Jacobian eigenvalue as an early warning signal for critical transitions. To illustrate this approach, we apply it to ecological meta-foodweb models, which are strongly nonlinear dynamical multi-layer networks. Our analysis shows that accurate results can be obtained, although the data demand of the method is still high.

# 1. Introduction

As humans we are dependent on the functioning of complex systems on many scales, ranging from our own body with its interlinked metabolic, signalling and microbial networks, via technical and organizational systems such as power grids and political systems, to the planetary-scale supply chains and the climate systems. All of these are complex nonlinear many-variable systems, and as such at a risk of undergoing sudden, qualitative and potentially irreversible transitions [1]. Over the past decades, there has been a steadily growing interest in methods that provide early warning of such transitions [2,3]. In particular, there is a growing awareness that such methods are needed for systems that are spatial or network based and hence inherently high-dimensional [4–6].

The traditional approach to anticipating qualitative transitions in complex systems is mechanistic modelling. Many well-understood systems can be modelled so precisely that the model can accurately predict the threshold parameter values where transitions occur. For technical applications such as aircraft flight or stability of structures, the model-based stability analysis is well established and in many cases part of a legally mandated licensing process [7]. However, model-based approaches tend to produce poor results in systems that are less well known as seemingly minor details of the model can sometimes drastically affect transition points.

It has long been known that critical transitions are generally preceded by critical slowing down [8,9]. This phenomenon leads to a distinctive increase in the auto-correlation and cross-correlations of time series before a transition. The advantage of correlation-based warning signs is that detailed understanding of the system is not needed. Its disadvantage is that high-frequency time-series data is necessary for robust warning, which imposes a strong, and for many applications prohibitive, constraints.

The model-based and correlation-based approaches to predicting critical transitions can be seen as two extreme strategies. The former is entirely based on structural knowledge of the system and uses real-world data at most to fit model parameters. The latter eliminates the need for structural information, at the cost of a high time-series data demand [10,11].

In many potential applications, there are aspects of the system that are well understood because different variables are related via physical laws or subject to logical constraints [11]. If such bits of 'structural' knowledge are available in a system it is desirable to exploit them for the construction of early warning signals. However, the same systems may also contain aspects that are considerably less well understood and hence make purely model-based predictions unreliable. This defines the need for a middle way, where available structural information on a system is used to reduce the data demand, while time-series data is used to close gaps in the understanding in areas where structural information is not available [2].

A common approach to finding a middle way in the construction of early warning signs is to use data assimilation approaches to fit and/or continuously improve a dynamical model [11–13]. A classical application of these approaches is the eutrophication of shallow lakes [14,15], for which good early warning signs can be constructed using techniques such as Bayesian learning and Kalman filters [16–18]. However, a broad investigation of such approaches shows that they may result in warning signs that are 'faint and late' [11] and hence Boettiger and Hastings point out advantages of a model-based approach [15].

A different approach to critical transitions is based more directly on linear stability analysis [19,20]. In [19], the authors formulated a model that was sufficiently general to encompass the whole class of conceivable models into which a given system could potentially fall. Using the so-called generalized modelling approach, the Jacobian matrices were computed that govern the system's response to perturbations and from which the bifurcation points can be computed. Because the underlying models contain unknown functional relationships, the Jacobian matrices that are thus obtained still contain unknown parameters. The authors used time-series data to eliminate these remaining uncertainties. They showed that in this way accurate early warning signals can be constructed that only require very limited time-series data.

To understand the scope of the challenge, it is useful to recall that inferring a system's Jacobian matrix from the covariance of variables, means constructing causality from correlation. Several approaches have been proposed to tackle this widely recognized challenge. A simple ad-hoc approach is the so-called Granger causality [21] which is based on the idea that causality causes correlation with a detectable time delay. A more sophisticated variant of this idea, that resolves some of its shortcomings by a clever use of Taken's theorem [22] was proposed by Sugihara *et al.* [23].

The present paper takes a different approach to Jacobian reconstruction by building instead on a fundamental theorem on stochastic processes. In a system that is subject to some noise, the cross correlations in timeseries encode very similar information to the Jacobian matrix of its deterministic backbone. Considered in isolation the cross correlations are not sufficient to reconstruct the full Jacobian; however, a complete reconstruction is possible if we have some additional structural information. In systems that can be described as complex networks the network structure imposes constraints on which variables can interact directly, which in turn implies that some entries of the Jacobian must be zero. In a sufficiently sparse network, and particularly in multi-layer networks, knowing these zeros provides sufficient information to reconstruct the remainder of the Jacobian from time series correlations.

For illustration, we apply Jacobian reconstruction approach to an ecological meta-foodweb model, formulated as a dynamical multi-layer network. By comparing with the known ground truth of the model, i.e. its exact Jacobian, we show that the Jacobian can be reconstructed faithfully and demonstrate its value as an early warning signal. We find that despite leveraging the structural information, the amount of time-series data required for accurate results is at present still prohibitively high for the ecological application. However, we discuss several avenues of future research that may reduce the data requirements to a point where the method becomes widely applicable.

## 2. Mathematical background

In every dynamical system that is in the vicinity of some form of stationary long-term behaviour the response of the system to small perturbations in the variables can be captured by some matrix. In the simplest case, the system is a system of ordinary differential equations (ODEs)

$$\dot{x} = f(x, p) \tag{2.1}$$

in which a vector of variables $x$ evolves in time in a way that is dependent on a set of parameters $p$. The simplest form of long-term behaviour is rest in a stationary state $x^*(p)$, such that

$$f(x^*, p) = 0. \tag{2.2}$$

The response to sufficiently small perturbations in the steady state is then described by the Jacobian matrix $\mathbf{J}$, whose elements are computed as

$$J_{ij} = \frac{\partial}{\partial x_j} \dot{x}_i \Big|_{x=x^*}. \tag{2.3}$$

The steady state under consideration is stable when all eigenvalues of the Jacobian have negative real parts. A smooth change in the parameters $p$ will generally cause the eigenvalues to change smoothly. When such a change causes one or more eigenvalues to acquire positive real parts, then a bifurcation occurs in which the dynamics change qualitatively.

Because the Jacobian is a real matrix its eigenvalues are real or form complex conjugate pairs. Hence there are two fundamental types of bifurcations. In bifurcations of fold-type a single real eigenvalue acquires a positive real part as it passes through zero. Several different forms of this bifurcation are commonly encountered in dynamical systems (fold or saddle-node bifurcation, pitchfork bifurcation, transcritical bifurcation), but generally speaking these are associated with a change in the number of steady states in the system or the exchange of stability properties.

In a Hopf bifurcation, a complex conjugate eigenvalue pair acquires a positive real part by crossing the imaginary axis of the complex plane. This bifurcation is generally associated with the onset of at least transient oscillations as the system departs from the steady state.

Both types of bifurcations can occur in several different forms, some of which cause only continuous and hence non-catastrophic transitions (say, replacing stationary behaviour with low amplitude oscillations), while others lead to a catastrophic (i.e. discontinuous, hard-to-reverse) departure from the previous state.

One can distinguish between different types of bifurcations by computing normal form parameters that are functions of higher derivatives of $f$. However, this is beyond the scope of the present paper.

In spatially extended systems, which are defined on a continuous space or on a spatial network of discrete nodes, the fundamental bifurcations can come in two flavours: a bifurcation may affect all points in space simultaneously in the same way, or it may affect points in space differently leading to the formation of spatial patterns. For clarity, the bifurcations of the latter type are called Turing bifurcations (fold-like case) and wave instabilities (Hopf-like case).

While not every bifurcation in a dynamical system is a critical transition, any bifurcation occurring in an important real-world system is certainly a cause for concern. In this spirit, our aim in the following is to reconstruct the Jacobian of a dynamical system from data in order to determine its leading eigenvalue. If the real part of this eigenvalue approaches zero, we interpret this as a warning signal for an impending bifurcation.

## 3. Jacobian reconstruction

Our aim in this section is to formulate a method that can reconstruct the Jacobian matrix of a dynamical system from time series. We do this by expanding on the work of Honerkamp [24], van Kampen [25] and Steuer et al. [26].

We start from a stochastic time series that fluctuates around a steady-state $x^*$ of the underlying deterministic backbone of the system. As shown in [24–26] and reproduced in the electronic supplementary material the Jacobian $\mathbf{J}$ close to $x^*$ is related to the covariance matrix $\boldsymbol{\Gamma}$ of time series, and to the fluctuation matrix $\mathbf{D}$ of the noise, via the equation

$$\mathbf{J}\boldsymbol{\Gamma} + \boldsymbol{\Gamma}\mathbf{J}^{\mathrm{T}} = -2\mathbf{D}. \tag{3.1}$$

Consider that a Jacobian matrix of linear dimension $N$ contains $N^2$ independent entries. By contrast, equation (3.1) equates two symmetric matrices, and hence imposes only $N(N+1)/2$ constraints on the elements of $\mathbf{J}$. Therefore, in any application with multiple variables ($N > 1$) the system is underdetermined such that we cannot recover the complete Jacobian purely from the time series.

Recovery of the full Jacobian may be possible if we can leverage some additional information on the system. Fortunately, in many applications some structural information is easily accessible. In particular in large complex spatial systems or reasonably sparse networks, we know that certain variables cannot interact and hence the corresponding elements of the Jacobian must be zero.

The structural zeros yield a set of constraints of the form $J_{ij} = 0$. If we can identify at least $N(N-1)/2$ such zero entries, then the time series contain enough information to reconstruct all remaining Jacobian entries.

Particularly in large networks such structural zeros are common due to informational constraints. In many examples of dynamics on networks only neighbouring nodes are interacting hence Jacobian elements corresponding to interactions between non-adjacent nodes must be zero. In a network of mean degree $z$, this results in the necessary number of $N(N-1)/2$ zeros if $N \geq 2z - 1$. For example for a mean degree of $z = 3$ every network with at least five nodes meets this condition.

In multilayer networks, additional informational constraints typically exist that make the condition even easier to satisfy. Below we consider an example where $S$ different species occupy

$P$ habitat patches in a landscape. Within a patch, species can coexist such that the dynamical dimension of the system is $N = SP$ dynamical variables and hence $SP(SP - 1)/2$ zeros are needed. An individual of a given species can affect the population of the same species in a different patch by migrating there. It can also affect the dynamics of other species at the patch where it is currently at, e.g. by predation. However, it cannot affect the dynamics of a different species in a different patch, which implies $SP(S - 1)(P - 1)$ zeros in the Jacobian. This means that even if the geographical network is fully connected then sufficient structural zeros are available if $2(S + P) < SP - 3$. For example, for $S = 3$ species even fully connected networks will have enough zeros if there are at least six patches. Again, the condition becomes easier to satisfy for a higher number of species and/or patches.

Given a time series of the system's $N$ variables and a set of $G$ additional structural constraints, with $G \geq N(N - 1)/2$ we now solve the equation (3.1) to estimate the non-zero entries of the Jacobian. To understand how this equation is solved let us first consider the two-dimensional example

$$\begin{pmatrix} J_{11} & J_{12} \\ J_{21} & J_{22} \end{pmatrix} \begin{pmatrix} \Gamma_{11} & \Gamma_{12} \\ \Gamma_{12} & \Gamma_{22} \end{pmatrix} + \begin{pmatrix} \Gamma_{11} & \Gamma_{12} \\ \Gamma_{12} & \Gamma_{22} \end{pmatrix} \begin{pmatrix} J_{11} & J_{21} \\ J_{12} & J_{22} \end{pmatrix} = -2 \begin{pmatrix} D_{11} & D_{12} \\ D_{12} & D_{22} \end{pmatrix}$$

which implies the independent conditions

$$2J_{11}\Gamma_{11} + 2J_{12}\Gamma_{12} = -2D_{11} \tag{3.2}$$

$$J_{11}\Gamma_{12} + J_{12}\Gamma_{22} + J_{21}\Gamma_{11} + J_{22}\Gamma_{12} = -2D_{12} \tag{3.3}$$

and
$$2J_{21}\Gamma_{12} + 2J_{22}\Gamma_{22} = -2D_{22} \tag{3.4}$$

with the second condition applying to both off-diagonal terms. The left-hand side of these equations is a linear system. Hence we can write the conditions in the form

$$\mathbf{B}j = -2d \tag{3.5}$$

where $\mathbf{B}$ is a matrix, $j$ is a column vector that contains the entries of the Jacobian, i.e. $j = (J_{11}, J_{21}, J_{12}, J_{22})^{\mathrm{T}}$ and $d$ is the corresponding vector for $\mathbf{D}$. For the two-dimensional example, this reads

$$\begin{pmatrix} 2\Gamma_{11} & 0 & 2\Gamma_{12} & 0 \\ \Gamma_{12} & \Gamma_{11} & \Gamma_{22} & \Gamma_{12} \\ \Gamma_{12} & \Gamma_{11} & \Gamma_{22} & \Gamma_{12} \\ 0 & 2\Gamma_{12} & 0 & 2\Gamma_{22} \end{pmatrix} \begin{pmatrix} J_{11} \\ J_{21} \\ J_{12} \\ J_{22} \end{pmatrix} = -2 \begin{pmatrix} D_{11} \\ D_{12} \\ D_{12} \\ D_{22} \end{pmatrix} \tag{3.6}$$

The form of this equation suggests that we can solve it for $j$ by multiplying $\mathbf{B}^{-1}$ from the left. However, we have to take care because $\mathbf{B}$ is not invertible because the two centre rows are identical, which is a consequence of the missing information. We can fix this problem by using the additional constraints. Imposing structural constraints on the two-variable system makes this example almost pointless, but, for the purpose of illustration, let us assume that we know that variable 1 cannot depend on variable 2 and hence $J_{12} = 0$. We can represent this constraint in the same matrix equation as the system by adding it as an additional line,

$$\begin{pmatrix} 2\Gamma_{11} & 0 & 2\Gamma_{12} & 0 \\ \Gamma_{12} & \Gamma_{11} & \Gamma_{22} & \Gamma_{12} \\ \Gamma_{12} & \Gamma_{11} & \Gamma_{22} & \Gamma_{12} \\ 0 & 2\Gamma_{12} & 0 & 2\Gamma_{22} \\ 0 & 0 & 1 & 0 \end{pmatrix} \begin{pmatrix} J_{11} \\ J_{21} \\ J_{12} \\ J_{22} \end{pmatrix} = -2 \begin{pmatrix} D_{11} \\ D_{12} \\ D_{12} \\ D_{22} \\ 0 \end{pmatrix} \tag{3.7}$$

If this is the only constraint then we can now drop the third row of the matrix and the third entry of the vector on the right-hand side and solve for $j$ by matrix inversion. In practice there are typically additional constraints and hence the system will be overdetermined. In this case we use least squares optimization to find an approximate solution, which can be done in closed form, so no numerics are necessary. As we will see below the form of equation (3.7) is very convenient

for finding the least squares solution. In particular it allows us to obtain a analytical closed-form solution for $j$.

Let us now generalize from the two-dimensional example to systems with an arbitrary number of variables. For this purpose we define the vectorization operator [27]

$$\text{vec}(\mathbf{X}) = (X_{11}, X_{21}, \ldots, X_{N1}, X_{12,} \ldots)^{\mathrm{T}}. \tag{3.8}$$

We can now write equation (3.5) as

$$\mathbf{B}\,\text{vec}(\mathbf{J}) = -2\,\text{vec}(\mathbf{D}), \tag{3.9}$$

i.e. the vectorization of a matrix is the columns of the matrix stacked on top of each other. To find the general form of $\mathbf{B}$, we start from equation (3.1) and vectorize both sides, which yields

$$\text{vec}(\mathbf{J}\boldsymbol{\Gamma} + \boldsymbol{\Gamma}\mathbf{J}^{\mathrm{T}}) = \text{vec}(-2\mathbf{D}). \tag{3.10}$$

Because vectorization is a linear operator we can pull the $-2$ out of the vectorization on the right-hand side and apply the vectorization separately to the two terms on the left-hand side, hence

$$\text{vec}(\mathbf{J}\boldsymbol{\Gamma}) + \text{vec}(\boldsymbol{\Gamma}\mathbf{J}^{\mathrm{T}}) = -2\,\text{vec}(\mathbf{D}). \tag{3.11}$$

It is known [27] that for matrices $\mathbf{X}, \mathbf{Y}, \mathbf{Z}$ the following identity holds (see electronic supplementary material):

$$\text{vec}(\mathbf{XYZ}) = (\mathbf{Z}^{\mathrm{T}} \otimes \mathbf{X})\,\text{vec}(\mathbf{Y}), \tag{3.12}$$

where $\otimes$ is the Kronecker product of matrices defined by

$$\mathbf{X} \otimes \mathbf{Y} = \begin{pmatrix} X_{11}\mathbf{Y} & X_{12}\mathbf{Y} & \ldots \\ X_{21}\mathbf{Y} & X_{22}\mathbf{Y} & \ldots \\ \vdots & \vdots & \ddots \end{pmatrix} \tag{3.13}$$

We now substitute $\mathbf{Y} = \mathbf{J}, \mathbf{Z} = \boldsymbol{\Gamma}$ and $\mathbf{X} = \mathbf{I}$, where $\mathbf{I}$ is the identity matrix of appropriate size. This yields

$$\text{vec}(\mathbf{J}\boldsymbol{\Gamma}) = \text{vec}(\mathbf{IJ}\boldsymbol{\Gamma}) = (\boldsymbol{\Gamma} \otimes \mathbf{I})\,\text{vec}(\mathbf{J}) \tag{3.14}$$

which brings the first term from equation (3.11) into the desired form. If we try the same for the second term we find

$$\text{vec}(\boldsymbol{\Gamma}\mathbf{J}^{\mathrm{T}}) = \text{vec}(\boldsymbol{\Gamma}\mathbf{J}^{\mathrm{T}}\mathbf{I}) = (\mathbf{I} \otimes \boldsymbol{\Gamma})\,\text{vec}(\mathbf{J}^{\mathrm{T}}) \tag{3.15}$$

which is not quite the desired form because vectorization of the transpose of $\mathbf{J}$ appears rather than the vectorization of $\mathbf{J}$ itself. The vectorization of a matrix and the vectorization of its transpose are not identical but closely related. Consider that for our two-dimensional the two vectorizations are related by

$$\text{vec}(\mathbf{J}^{\mathrm{T}}) = \begin{pmatrix} J_{11} \\ J_{12} \\ J_{21} \\ J_{22} \end{pmatrix} = \mathbf{C} \begin{pmatrix} J_{11} \\ J_{21} \\ J_{12} \\ J_{22} \end{pmatrix} = \mathbf{C}\,\text{vec}(\mathbf{J}), \tag{3.16}$$

where

$$\mathbf{C} = \begin{pmatrix} 1 & 0 & 0 & 0 \\ 0 & 0 & 1 & 0 \\ 0 & 1 & 0 & 0 \\ 0 & 0 & 0 & 1 \end{pmatrix} \tag{3.17}$$

is a permutation matrix. In the general case, we can still write

$$\text{vec}(\mathbf{J}^{\mathrm{T}}) = \mathbf{C}\,\text{vec}(\mathbf{J}) \tag{3.18}$$

where $\mathbf{C}$ is now a permutation matrix of size $N^2 \times N^2$, one can construct this matrix from blocks of size $N \times N$ as

$$\mathbf{C} = \begin{pmatrix} \mathbf{C}_{11} & \mathbf{C}_{12} & \cdots \\ \mathbf{C}_{21} & \mathbf{C}_{22} & \cdots \\ \vdots & \vdots & \ddots \end{pmatrix} \tag{3.19}$$

Here, $\mathbf{C_{nm}}$ is a $N \times N$-matrix defined by

$$(\mathbf{C_{nm}})_{ij} = \delta_{im}\delta_{nj} \tag{3.20}$$

where $\delta$ is the Kronecker delta.

Using the commutation matrix we can write equation (3.15) as

$$\mathrm{vec}(\boldsymbol{\Gamma}\mathbf{J}^{\mathsf{T}}) = (\mathbf{I} \otimes \boldsymbol{\Gamma})\,\mathrm{vec}(\mathbf{J}^{\mathsf{T}}) = (\mathbf{I} \otimes \boldsymbol{\Gamma})\,\mathbf{C}\,\mathrm{vec}(\mathbf{J}) \tag{3.21}$$

Substituting this relationship and equation (3.14) into equation (3.11) we get the desired form

$$\underbrace{((\boldsymbol{\Gamma} \otimes \mathbf{I}) + (\mathbf{I} \otimes \boldsymbol{\Gamma})\,\mathbf{C})}_{\mathbf{B}}\underbrace{\mathrm{vec}(\mathbf{J})}_{j} = -2\underbrace{\mathrm{vec}(\mathbf{D})}_{d}. \tag{3.22}$$

We now know how to construct the matrix $\mathbf{B}$ such that we can write the system in the form

$$\mathbf{B}j = -2d. \tag{3.23}$$

However, this system is still underdetermined. If we know that some elements of $j$ must be zero we can represent this knowledge in a matrix $\mathbf{U}$. For this purpose, we can gather the respective elements of $j$ in an ordered list $\mathcal{U}$ and then define $\mathbf{U}$ as an $|\mathcal{U}| \times N$ matrix with

$$U_{nm} = \begin{cases} 1 & \text{if } j_m \text{ is the } n\text{-th entry of } \mathcal{U} \\ 0 & \text{otherwise} \end{cases} \tag{3.24}$$

Each row of this matrix represents one of the constraints that we wish to impose. For example if in the fourth row the only non-zero entry is in the eighth column this means our fourth condition is that the eighth element of $j$ must be zero.

In analogy to the small example, we can now impose the additional conditions on the system by stacking them below the matrix $\mathbf{B}$ such that equation (3.23) becomes

$$\begin{pmatrix} \mathbf{B} \\ \mathbf{U} \end{pmatrix} j = -2 \begin{pmatrix} d \\ 0 \end{pmatrix} \tag{3.25}$$

where $0$ is a column vector containing $|\mathcal{U}|$ zeros. The () that appear in this equation should be read as a block-wise notation for matrices/vectors, where rows are stacked on top of each other in analogy to equation (3.7).

To simplify the notation we introduce

$$\hat{\mathbf{B}} = \begin{pmatrix} \mathbf{B} \\ \mathbf{U} \end{pmatrix} \quad \hat{d} = \begin{pmatrix} d \\ 0 \end{pmatrix} \tag{3.26}$$

which allows us to write the whole set of conditions once again in the form

$$\hat{\mathbf{B}}j = \hat{d} \tag{3.27}$$

In practice, this will now be an overdetermined system, such that no exact solution exists. However, finding an approximation that minimizes the squares of the deviations in each row is a well-known problem. The known solution [28,29] (cf. electronic supplementary material) for this problem is

$$j = (\hat{\mathbf{B}}^{\mathsf{T}}\hat{\mathbf{B}})^{-1}\hat{\mathbf{B}}^{\mathsf{T}}\hat{d} \tag{3.28}$$

where the expression $(\hat{\mathbf{B}}^{\mathsf{T}}\hat{\mathbf{B}})^{-1}\hat{\mathbf{B}}^{\mathsf{T}}$, the pseudo-inverse of $\hat{\mathbf{B}}$, appears.

The equation provides a closed-form solution by which the Jacobian elements can be computed from the covariance matrix of the time series, a known or estimated fluctuation matrix, and a set of additional structural constraints on the Jacobian.

Typically the least-squares fit will not set the Jacobian elements governed by the structural constraints exactly to zero. One may, therefore, enforce these known zeros by setting the respective elements of the Jacobian explicitly to zero after the computation of equation (3.28) has finished. In numerical experiments, described below, we found that this improved the accuracy of Jacobian eigenvalues estimated by this method.

To summarize, the Jacobian $\mathbf{J}$ of an $N$-dimensional system close to a steady state can be reconstructed as follows:

(i) Compute the co-variance matrix from the time-series data, $\Gamma_{ij} = \langle X_i X_j \rangle$.
(ii) Construct the diagonal fluctuations matrix $\mathbf{D}$, and compute $d = \text{vec}(\mathbf{D})$. In some systems these fluctuations can be measured directly, otherwise a reasonable approximation may be derived based on assumptions on the underlying noise process [25].
(iii) Construct the permutation matrix $\mathbf{C}$ according to equations (3.17) and (3.18).
(iv) Compute the matrix
$$\mathbf{B} = (\Gamma \otimes \mathbf{I} + (\mathbf{I} \otimes \Gamma)\,\mathbf{C}),$$
where $\mathbf{I}$ is the $N \times N$ identity matrix.
(v) Define $j = \text{vec}(\mathbf{J})$ and identify at least $N(N-1)/2$ elements of $j$ that must be zero due to structural constraints. Use these to construct the matrix $\mathbf{U}$ (see equation (3.24)). The column-dimension of $\mathbf{U}$ is the row dimension of $j$, and row dimension of $\mathbf{U}$ is identical to the number of structural constraints. The matrix has exactly one non-zero entry in each row such that $U_{nm} = 1$ if the $n$th structural condition reads $j_m = 0$.
(vi) Construct
$$\hat{\mathbf{B}} = \begin{pmatrix} \mathbf{B} \\ \mathbf{U} \end{pmatrix} \quad \hat{d} = \begin{pmatrix} d \\ 0 \end{pmatrix}$$
(vii) Compute
$$j = (\hat{\mathbf{B}}^{\mathrm{T}} \hat{\mathbf{B}})^{-1} \hat{\mathbf{B}}^{\mathrm{T}} \hat{d}$$
and recover the Jacobian $\mathbf{J}$ from its vectorization $j$.
(viii) Set the elements of $\mathbf{J}$ governed by structural constraints explicitly to zero.

## 4. Application to a meta-foodweb model

In the following, we explore if the Jacobian matrix can be reconstructed sufficiently accurately to warn of impending critical transitions. We constructed a test system of realistic complexity for which the Jacobian matrix is nevertheless known analytically, and then created noisy time series for this system.

In the simulations we used a meta-foodweb model already studied in [30,31] (see electronic supplementary material for details). The model consists of a spatial network of $P$ habitat patches linked by avenues of species dispersal (figure 1). Each patch harbours a complex foodweb, consisting of $S$ nodes that represent populations of different species, which are linked by predator–prey interactions. The dynamics of the system are given by a set of differential equations that govern the changes in variables due to diffusive dispersal between patches and biological processes occurring within a patch (primary production, predator–prey interaction, natural mortality).

The model contains several complicating factors encountered in real systems. The functions used to model the biological processes are strongly nonlinear. They capture various saturation effects and realistic responses to the availability of different food sources (prey switching). The dynamics of different species occurs on different time scales according to biological scaling relationships which relate a species position in the foodweb to its expected biomass turnover rate

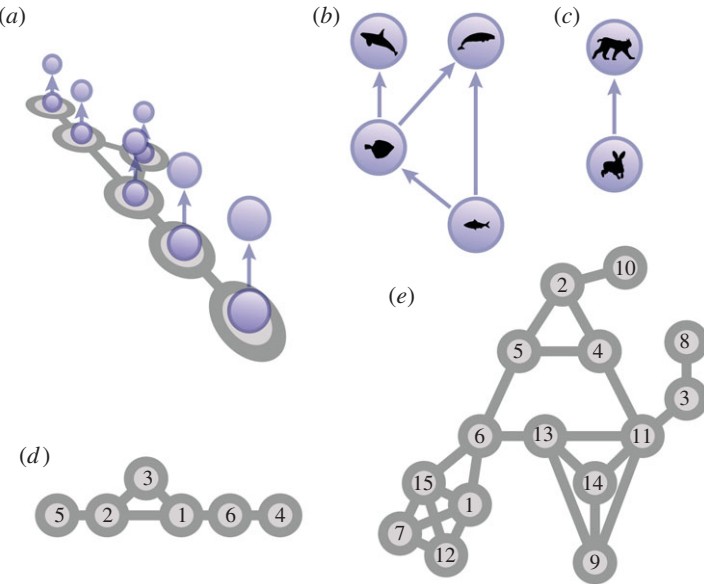

**Figure 1.** Schematic of the model system. The system is a multilayer network, where copies of an ecological food web exist in different geographical patches (*a*). We consider an intra-guild predation food web with an additional predator (*b*) and a predator–prey system (*c*). For the spatial network, we use the smallest completely asymmetric graph (*d*) and a larger random topology, generated as a random geometric graph (*e*). (Online version in colour.)

[32]. Similar scaling relationships also govern the rate at which different species disperse across the spatial network [30].

We consider two different versions of the patch topology. The smaller of the two consists six patches which are connected in such a way that they form the smallest completely asymmetric network (figure 1*d*). The larger one contains 15 patches and was generated as a random geometric graph (figure 1*e*). It has a comparatively large diameter and high clustering and contains several symmetries that are characteristic for this type of spatial networks [33]. For the food webs, we used a predator–prey system consisting of two species (figure 1*c*), as well as a four-species system in the shape of the so-called intra-guild predation motif with an additional predator (figure 1*b*), a common foodweb motif.

Work by Nakao & Mikhailov [34] and the extension of their approach to meta-foodwebs [30] showed that network models behave analogously to dynamical systems in continuous space. Hence the theory of pattern formation in partial-differential equation systems can transferred almost exactly to these dynamical networks. This means that our test systems can exhibit instabilities that are best described as pattern-forming bifurcations, specifically Turing and Wave instabilities. Pattern forming instabilities in predator-prey systems in continuous space were studied in detail in [35] which made it easy to locate these bifurcations also in our multi-layer network predator–prey system (figure 2).

Using a generalized modelling approach [36–38], we analytically computed the Jacobian that describes the two example food webs on arbitrary spatial topologies. We then picked specific realizations of models in which relevant bifurcations occurred. To produce noisy time series, we simulated these models with added noise using the Euler–Maruyama method (see electronic supplementary material) [39].

## 5. Results

As a first test we generated simulated noisy time series containing $2 \times 10^5$ data points from the two-species system on the six-patch topology. Parameter values for these simulations were

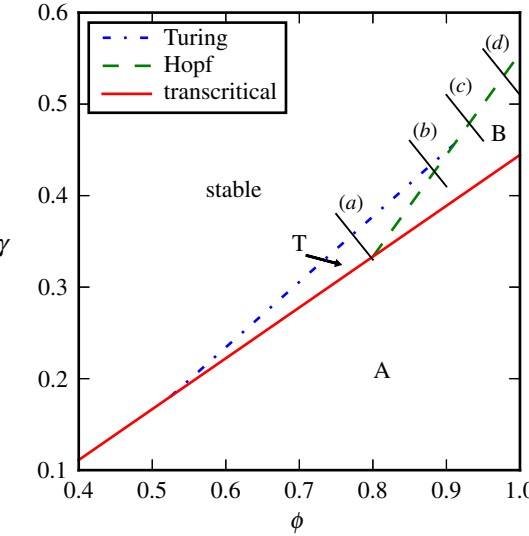

**Figure 2.** Bifurcation diagram of the predator-prey system on the six patches topology. Stability of the state under consideration changes in response to changes in the sensitivity of biomass production to producer biomass $\phi$ and the sensitivity of predation to prey biomass $\gamma$. The state under consideration is stable in the top left area. Stability is lost when either of three bifurcations occur (Turing, transcritical, Hopf). After the loss of stability the system approaches a state of homogeneous oscillations (A), a different homogeneous stationary state (B), or a state of stationary patterns (T). The bifurcation diagram was computed using the master stability function approach from [30] (see electronic supplementary material). It corresponds directly to Fig. 1 from [35] which studies a predator-prey system in continuous space. Lines (a–d) indicate the transects used for the corresponding simulations in figure 3. (Online version in colour.)

chosen to lie on four transects through the parameters space that crossed bifurcation points. For each of these time series, we then reconstructed the Jacobian matrix and computed the leading eigenvalue. Before the bifurcation point the estimated eigenvalues are in very good agreement with the known ground truth provided by the analytic eigenvalues (figure 3).

As we follow the transects the real part of the leading eigenvalue crosses zero in a bifurcation. When the bifurcation occurs the reconstructed eigenvalue departs from the analytical value. This behaviour is expected as the analytical solution continues to show the eigenvalues around the, now unstable, steady state, whereas the reconstruction algorithm computes the leading eigenvalue associated with the new dynamics, which has now departed from the previous steady state.

We note that in transect (d) the reconstructed eigenvalue is almost exactly zero in a wide region after the bifurcation. This happens because the system approaches a stable limit cycle for which the leading Lyapunov exponent is zero. The recovery of this zero by the algorithm provides some (unexpected) evidence suggesting that the method reveals some salient information even in non-stationary states. To test whether the full Jacobian rather than just the leading eigenvalue is reconstructed correctly we compare the full reconstructed eigenvalue spectrum to the analytical ground truth (figure 4a). The results show that in stable steady states the relative accuracy for the reconstruction of all eigenvalues is comparable, which is the expected behaviour.

After destabilization the simulated system departs from the analytical steady state and thus discrepancies appear that are not due to reconstruction errors but arise as we are now comparing different states. It is interesting, but perhaps not surprising, that this discrepancy is larger for the eigenvalues close to zero. Whereas very negative eigenvalues that are likely grounded in strong restorative forces are relatively unaffected.

Careful examination of the transects (figure 3) suggests that the accuracy of reconstruction improves as we approach the bifurcation point. To explore this further, we generated 15 transects

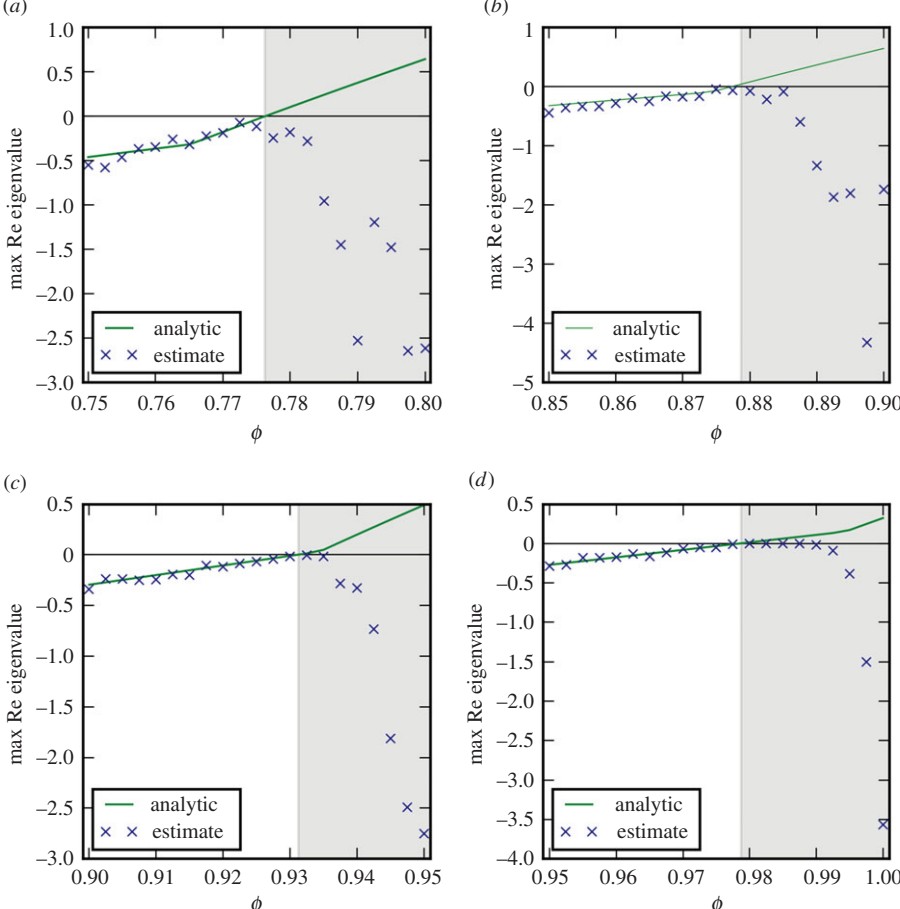

**Figure 3.** Comparison of the analytical ground truth for the leading eigenvalue with an eigenvalue estimate from Jacobian reconstruction. The panels correspond to the four transects shown in figure 2. Estimates are in good agreement with the analytical value until system parameter $\phi$ crosses a threshold where stability is lost and the time series depart from the steady state (shaded region). (Parameters $\phi$, $\gamma$: (a) 0.75, 0.38 to 0.8, 0.33; (b) 0.85, 0.46 to 0.9, 0.41; (c) 0.9, 0.51 to 0.95, 0.46; (d) 0.95, 0.56 to 1.0, 0.51. Bifurcations encountered are Turing (a,b) and Hopf (c,d)). (Online version in colour.)

in the vicinity of bifurcation points and generated three sets of time series along every one of the transects (see electronic supplementary material for details). The results confirm that the accuracy of the estimate improves as the bifurcation point is approached (figure 4b).

We now consider the case where parameters are slowly changing over a long simulation run. Our aim is to estimate the leading eigenvalue of the Jacobian over time as this slow change in the system is taking place. For this purpose, we apply the proposed method to reconstruct the Jacobian in sliding time window of length $\tau$. The results (figure 5) show that the estimates based on the sliding window are more noisy, undergoing visible fluctuations around the true value. Nevertheless, the trend of the eigenvalue approaching zero is still clearly captured.

To test the applicability to larger networks, we apply Jacobian reconstruction to a model system with four species on 15 patches (i.e. a 60-dimensional dynamical system). Even in this larger system, the accuracy is still quite good but the estimated eigenvalue is systematically slightly less than the true value (figure 6). We suspect that this may be the combined effect of the non-autonomous nature of this systems and the noise leading to bifurcation delay. This delay effect

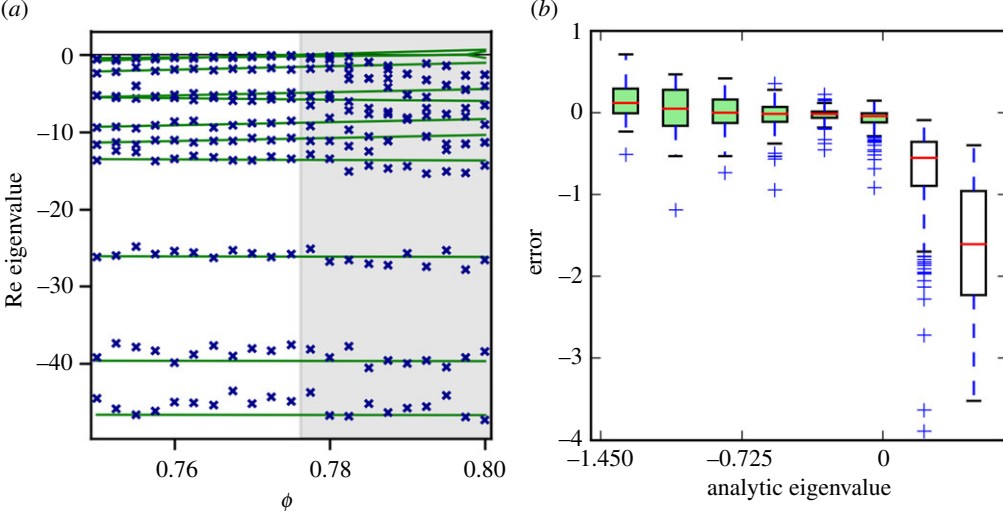

**Figure 4.** Accuracy of the eigenvalue estimation. (*a*) Comparison of analytic (lines) and reconstructed (crosses) eigenvalues shows that the reconstruction performs well in the stable region (white background) and yields comparable relative error for all eigenvalues. Once stability is lost (shaded region) the system departs from the analytical steady state, hence a discrepancy in the eigenvalues appears. (Parameters as in figure 3*a*). (*b*) For a closer analysis of the accuracy of the leading eigenvalue, we repeated the analysis of figure 3 three times and plot the deviation of the reconstructed eigenvalue depending on the analytic eigenvalue. The accuracy approves significantly as we approach the bifurcation point. After destabilization large errors appear that are due to the departure from the benchmark state, rather than inaccuracies in the reconstruction. (Online version in colour.)

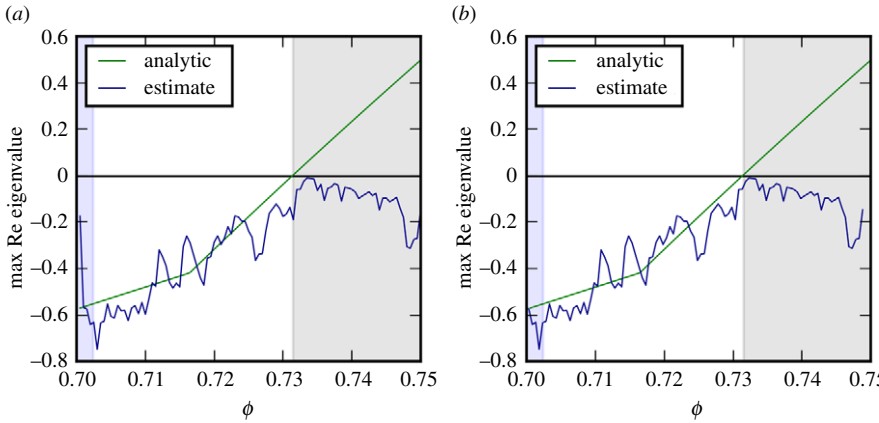

**Figure 5.** Eigenvalue estimation in a system with continuously changing parameters. We compare the estimated eigenvalue from Jacobian reconstruction in a sliding window to the analytic ground truth for a predator–prey system on six geographical patches close to a Turing bifurcation. The panels show the same result, however the estimate is placed at the end of the observation window (*a*) or in the centre (*b*). For large $\phi$, the steady state described by the analytical eigenvalues is unstable (greyed region) and hence the reconstruction yields eigenvalues of a different state. The width of the sliding window is indicated on the left (shaded blue left edge). (Parameters: $\phi, \gamma$ is changed from 0.70, 0.35 to 0.75, 0.30). (Online version in colour.)

can be expected to be more pronounced in the larger food web due to the presence of higher-level predators whose dynamics happen on correspondingly longer time scales (leading to a wide range of time scales in the model). If this is the case then the reconstructed Jacobian eigenvalue

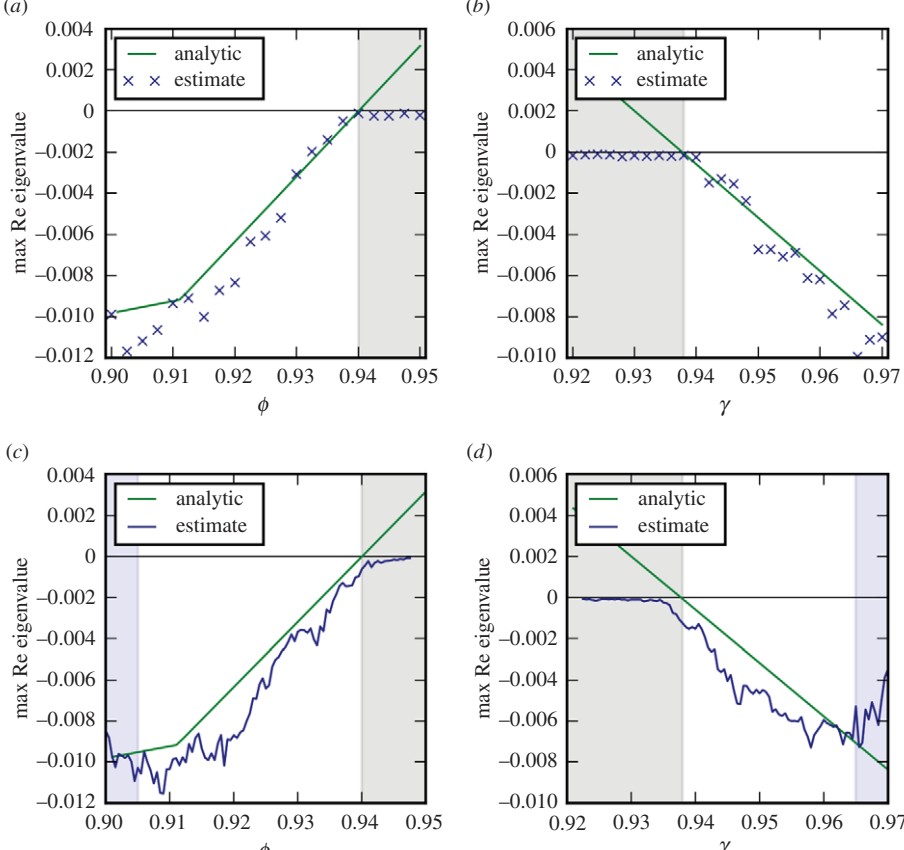

**Figure 6.** Eigenvalue estimations for the system with four species on 15 patches. We consider loss of stability as a result of the change of two different parameters, governing the nonlinearity of primary production, $\phi$ (*a,c*) and the nonlinearity of the functional response of attack rate to prey density, $\gamma$ (*b,d*). The leading eigenvalue of the Jacobian was reconstructed from time series for fixed parameters (*a,b*) and parameter transects (*c,d*). The steady state under consideration is unstable subsequent to a Hopf bifurcation (region shaded grey). For the transects estimates are shown in the centre of the sampling window (window width is indicated by blue area). See electronic supplementary material for parameters and details. (Online version in colour.)

may actually offer a better estimate of the relevant transition point than the analytical solution for the system without noise.

So far we have studied systems that were designed using the generalized modelling approach. We complement this by the analysis of a well-established ecological model, the Rosenzweig–MacArthur predator–prey model [40] with quadratic mortality and diffusion on the six-patch network (see electronic supplementary material for details). The results of Jacobian reconstruction (figure 7) show that the leading eigenvalue can be recovered with reasonable accuracy, again the accuracy of the estimate improves significantly as the system approaches the bifurcation point.

## 6. Summary and discussion

In this paper, we expanded on previous work by Honerkamp, van Kampen, Steuer and others to derive a closed-form expression for the reconstruction of Jacobian matrices from time-series data.

Our work contributes to the bigger overarching question how causality can be inferred from correlations. The calculation presented here shows that one can analytically infer causality (described by a Jacobian matrix) from correlations (covariance and fluctuation matrices) if some

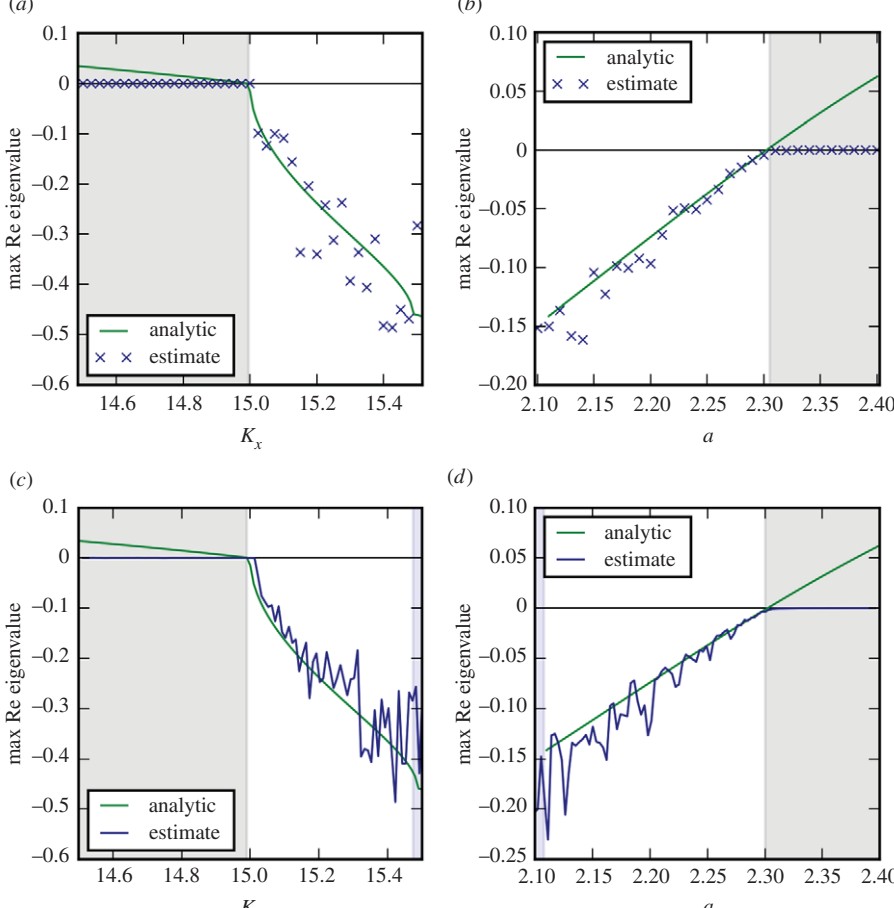

**Figure 7.** Eigenvalue estimations in the Rosenzweig–MacArthur with quadratic mortality and diffusion on the six-patch network. The steady state under consideration loses stability due to a Fold (*a,c*) and Hopf (*b,d*) bifurcations. The leading eigenvalue was reconstructed from simulation runs with fixed parameters (*a,b*) and slowly changing parameters (*c,d*). The accuracy of the estimates improves as the system approaches the bifurcation. (Estimated values are shown in the middle of the sampling window, indicated in blue. See electronic supplementary material for details). (Online version in colour.)

additional information is available. In this case, this additional information is the placements of some zeros of the Jacobian matrix. If the zeros and correlations are known exactly, then the causality can be inferred exactly. If this information is only known approximately, then the prediction will deteriorate. It can be expected that inaccuracies in the correlations should carry over almost linearly to the Jacobian, whereas the impact of misplaced zeros is more complicated and remains a question for future research.

For illustration, we applied the derived mathematical formula to the an ecological meta-foodweb model. This example illustrated that a relatively robust reconstruction of the leading eigenvalue of the Jacobian is possible even in a strongly nonlinear multi-layer network with dynamics on multiple time scales. However, the example also revealed that the required amount of data is still prohibitive for the ecological application.

There is reason to believe that future research and in particular a deeper mathematical understanding of the Jacobian reconstruction can significantly reduce the required amount of data. Particularly interesting in this respect is the observed increase in accuracy close to the bifurcation point. A promising goal for future exploration would be to understand how far this

region of heightened accuracy extends. If a real-world application under consideration is already close to bifurcation one might find that much less data is required to reach the desired accuracy.

Alternatively, it might be possible to reduce the data demand by optimizing the sampling scheme. One can envision an iterative scheme, similar to [41], where a small number of samples is used to find an initial estimate of the Jacobian. Using the initial estimate, one could then identify the relevant time scales and important entries in the covariance matrix and optimize the sampling effort accordingly.

A third alternative may be to use additional information that may be available. The advantage of our Jacobian-based approach is that it can take advantage of additional knowledge on the systems that may be available in ecological applications. For example, if predators do not interact predation and mortality rates should be linear, which fixes some entries of the Jacobian matrix of a generalized model. A previous study [19] suggests that the use of this information may reduce the data demand considerably. Meanwhile the method proposed here may be useful in fields where data are more readily available, such as studies of metabolism, power grids, or economic data. In the study of metabolism Jacobian reconstruction is already frequently used [42], for this application the present work yields an analytic closed-form solution to a problem that is so far solved by machine learning methods. For power grids, reconstructing Jacobians may be particularly interesting because it could yield deeper insights into the functioning of the system in addition to providing an early warning signal. For applications to economics, we note that the Jacobian is a representation of causality in the system. Closed-form Jacobian reconstruction thus offers a way to infer causality from correlation that is exact in the large data limit.

We note that by providing a closed-form analytical solution the proposed method differs qualitatively from other approaches. It thus lays the foundation for closed mathematical exploration where the Jacobian appears as part of a bigger problem. For example, one could explore the question how causality flows in certain epidemic models where the Jacobian correlation and noise are all explicit functions of the system state, but the state that the system approaches cannot be calculated analytically. In this case having another closed-form relationship between correlation, fluctuation and state available could allow to solve for the state and lead to a complete solution.

Finally, we note that the method is not limited to bifurcations of steady states but could be applied to Poincaré maps or stroboscopic projections to explore the dynamics of other attractors.

In summary, we find that Jacobian reconstruction is a promising approach to the analysis of complex systems near critical transitions, although the data requirements presently still limit its applicability. We expect that the closed-form solution derived in the present paper inspires future mathematical work to alleviate these requirements.

Data accessibility. This work did not produce new primary research data. Numerical codes used in the examples will be deposited at http://github.com/bridgewalker/JacRec.

Authors' contributions. T.G. proposed the work, E.B. derived the solution, E.B. and A.B. developed models and ran simulations, B.D. and T.G. supervised the work. All authors contributed to the manuscript.

Competing interests. We declare we have no competing interests.

Funding. This work was supported by Deutsche Forschungsgemeinschaft Research Unit FOR 1748, under grant contract no. Dr300/13-2 and by the EPSRC (EP/N034384/10). T.G. is supported by HIFMB a project of the Volkswagen Foundation (ZN3285) and the Ministry of Science and Culture of Lower Saxony.

Acknowledgements. The authors thank Caio Lucidius N. Azevedo and Nicolas Verschueren van Rees for valuable pointers to the algebra literature, and Raissa D'Souza for comments on an earlier version of the manuscript.

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
