## [Peer Review File · Proceedings. Mathematical, Physical, and Engineering Sciences]

Review History

RSPA-2019-0767.R0 (Original submission)

Review form: Referee 1

Is the manuscript an original and important contribution to its field?

Good

Is the paper of sufficient general interest?

Good

Is the overall quality of the paper suitable?

Good

Can the paper be shortened without overall detriment to the main message?

No

Do you think some of the material would be more appropriate as an electronic appendix?

Yes

Do you have any ethical concerns with this paper?

No

Recommendation?

Reject – article is not of sufficient interest (we will consider a transfer to another journal)

Comments to the Author(s)

The authors present an novel method for estimating Jacobians from time series data. To my eye the main methodological novelty, apart from some clever linear algebra, is the use of structural zeros in the estimation. However, it seems to me that the authors are fairly casual in their assumption that sufficient structural zeros can be known with certainty a priori. This is particularly relevant to ecological data where a) behaviorally mediated indirect effects are difficult to identify a priori and b) data are often sampled over fairly coarse time steps. As such, I think the authors need to rigorously evaluate the sensitivity of their approach to incorrect placement of structural zeros.

Secondarily, there are several extant approaches to inferring Jacobians from time series data (e.g. dynamic linear models, autoregressive models, etc). It seems relevant to ask how the new method performs compared to these alternatives.

Review form: Referee 2**Is the manuscript an original and important contribution to its field?**

Marginal

Is the paper of sufficient general interest?

Marginal

Is the overall quality of the paper suitable?

Acceptable

Can the paper be shortened without overall detriment to the main message?

Yes

Do you think some of the material would be more appropriate as an electronic appendix?

No

Do you have any ethical concerns with this paper?

No

Recommendation?

Major revision is needed (please make suggestions in comments)

Comments to the Author(s)

The aim of this manuscript is to propose a mixed technique half-based on some prior knowledge on the governing equations and half-based on data for predicting sudden, qualitative and potentially irreversible transitions which may occur in complex networks. When the network is sufficiently sparse, and knowing the topology of the network, the authors assess the non-zero elements of the Jacobian matrix from time series correlations to compute the leading eigenvalue of the so-obtained jacobian matrix to detect bifurcation of singular points. As expected with such an approach, the amount of data required for accurate results is at present still prohibitively high for any practical application to a network from the real world. Indeed, typically, this technique requires to compute correlation between time series of all related variables. Since each node is at least once linked with another one, this means that at least one variable is measured in each node. Moreover, to accurately capture correctly the dynamics, it should be also necessary to investigate

the node dynamics, meaning that even more variables should be measured. This is never clearly stated, but the present techniques applied to a N-dimensional system (here N means the number of variables spanning the state space of the full network) requires to measure the N variables which is a huge constraint for investigating a network. This is then combined with the knowledge of the adjacency matrix of the network.

What the authors are able to track is the occurrence of a bifurcation by detecting when the real part of the leading eigenvalue is close to zero. It is therefore erroneous to claim that there is a reconstruction of the jacobian matrix. This is only a technique for assessing the leading eigenvalue. There is no evidence in this paper that the full jacobian matrix is actually correctly estimated. Moreover, as the authors mentioned themselves, their technique can only assess the leading eigenvalue associated with an "attractor", that is, with the trajectory actually followed by the system. For instance, when a Hopf bifurcation occurs, after the bifurcation, they can only assess the leading eigenvalue of the limit cycle which results from the bifurcation. What would happen in a large network where a node has a high degree and for which it is known that the dynamics becomes more or less a noisy limit cycle? In that case, most likely, I suspect that the eigenvalue would be close to zero although no particular bifurcation is involved. The kind of bifurcation which can be tracked with this technique seems to be quite limited to strong bifurcation like those destabilizing a singular point. In fact, the very basic introduction to bifurcations inserted in this manuscript is limited to those observed in singular points. Would it be possible to extend the present technique to bifurcation on periodic orbit (the most frequently encountered in dynamical systems)?

Another question arises. Typically, the authors will be able to predict the destabilization of a stable singular point, but not its stabilization. This seems to be supported by Figs 6 and 7. This should be discussed.

The present technique is here tested for two very particular (strong) bifurcations arising on singular points. Typically, when a network would be on a stable singular point, there is no dynamics at all! So, to sum up, it is shown how a small network motif at rest could be destabilized. There is no indication that this technique will be still working with larger network and for nontrivial dynamics (I only mean here oscillating network). My feeling is that the amount of data is very large for the little gain discussed in this manuscript. Until an evidence on larger network (N = 100 for instance), it is rather hazardous to claim for any application to real network. The bifurcation of periodic orbits should be also investigated.

Other remarks

"some form of long-term behavior (stationarity, periodicity etc.)". Stationarity designates in the sense of Kolmogorov a behavior whose mean quantities are constant. Consequently, periodic or chaotic behaviors are stationary. Please clarify this sentence. In particular, what is here designated as a stationary state is in fact an equilibrium, a steady state, a fixed point or even a singular point in a more mathematical terminology.

I suppose that by "fold bifurcation" the authors mean saddle-node or tangent bifurcation. It should be clear that most of these bifurcations are also encountered on periodic orbits.

Typically, after a Hopf bifurcation (the common one), the oscillations are sustained since there is a limit cycle induced by such a bifurcation.

"[...] while others lead to a catastrophic (and potentially irreversible) departure from the steady state." Please provide here an example. In which sense is it catastrophic?

In general, I found the short introduction on bifurcations rather inaccurate.

The network considered in this paper are not “multilayer network” but “dynamical network” in which there is one variable per node. These dynamical networks are of a relatively low dimension ($N < 15$). This is a very strong limitation to this work.

Fig. 2. Trascritical → Transcritical

There is no evidence in this manuscript that the systems investigated have multiple timescales. How a system having a stable singular point for solution could have multiple timescales? Please explain.

“Particularly interesting in this respect is the observed increase in accuracy close to the bifurcation point.” Unclear. Please clarify.

Decision letter (RSPA-2019-0767.R0)

25-Feb-2020

Dear Dr Gross:

I am writing to inform you that your manuscript RSPA-2019-0767 entitled "A closed form for Jacobian reconstruction from timeseries and its application as an early warning signal in network dynamics" has been rejected in its present form for publication in Proceedings A.

The Editor has made this decision based on the advice of referees, and taking into account their own opinion of your paper. With this in mind we would like to invite a resubmission, provided the comments of the referees and any comments from the Editor are taken into account. This is not a provisional acceptance.

The resubmission will be treated as a new manuscript. Please note that resubmissions must be submitted within six months of the date of this email. In exceptional circumstances, extensions may be possible if agreed with the Editorial Office.

Please find below the comments made by the referees, not including confidential reports to the Editor, which I hope you will find useful. If you do choose to resubmit your manuscript, please include details of how you have responded to the comments, and the adjustments you have made.

Please note that we have a strict upper limit of 28 pages for each paper. Please endeavour to incorporate any revisions while keeping the paper within journal limits. Please note that page charges are made on all papers longer than 20 pages. If you cannot pay these charges you must reduce your paper to 20 pages before submitting your revision. Your paper has been ESTIMATED to be 16 pages. We cannot proceed with typesetting your paper without your agreement to meet page charges in full should the paper exceed 20 pages when typeset. If you have any questions, please do get in touch.

To upload a resubmitted manuscript, log into <http://mc.manuscriptcentral.com/prsa> and enter your Author Centre, where you will find your manuscript title listed under "Manuscripts with Decisions." Under "Actions," click on "Create a Resubmission." Please be sure to indicate that it is a resubmission, and ensure you enter this ID - RSPA-2019-0767 - as the previous submission number.

Yours sincerely
Raminder Shergill

proceedingsa@royalsociety.org

on behalf of
Professor G. Ambika
Board Member
Proceedings A

Reviewer(s)' Comments to Author:

Referee: 1

Comments to the Author(s)

The authors present an novel method for estimating Jacobians from time series data. To my eye the main methodological novelty, apart from some clever linear algebra, is the use of structural zeros in the estimation. However, it seems to me that the authors are fairly casual in their assumption that sufficient structural zeros can be known with certainty a priori. This is particularly relevant to ecological data where a) behaviorally mediated indirect effects are difficult to identify a priori and b) data are often sampled over fairly coarse time steps. As such, I think the authors need to rigorously evaluate the sensitivity of their approach to incorrect placement of structural zeros.

Secondarily, there are several extant approaches to inferring Jacobians from time series data (e.g. dynamic linear models, autoregressive models, etc). It seems relevant to ask how the new method performs compared to these alternatives.

Referee: 2

Comments to the Author(s)

The aim of this manuscript is to propose a mixed technique half-based on some prior knowledge on the governing equations and half-based on data for predicting sudden, qualitative and potentially irreversible transitions which may occur in complex networks. When the network is sufficiently sparse, and knowing the topology of the network, the authors assess the non-zero elements of the Jacobian matrix from time series correlations to compute the leading eigenvalue of the so-obtained jacobian matrix to detect bifurcation of singular points. As expected with such an approach, the amount of data required for accurate results is at present still prohibitively high for any practical application to a network from the real world. Indeed, typically, this technique requires to compute correlation between time series of all related variables. Since each node is at least once linked with another one, this means that at least one variable is measured in each node. Moreover, to accurately capture correctly the dynamics, it should be also necessary to investigate the node dynamics, meaning that even more variables should be measured. This is never clearly stated, but the present techniques applied to a N-dimensional system (here N means the number of variables spanning the state space of the full network) requires to measure the N variables which is a huge constraint for investigating a network. This is then combined with the knowledge of the adjacency matrix of the network.

What the authors are able to track is the occurrence of a bifurcation by detecting when the real part of the leading eigenvalue is close to zero. It is therefore erroneous to claim that there is a reconstruction of the jacobian matrix. This is only a technique for assessing the leading eigenvalue. There is no evidence in this paper that the full jacobian matrix is actually correctly estimated. Moreover, as the authors mentioned themselves, their technique can only assess the leading eigenvalue associated with an "attractor", that is, with the trajectory actually followed by the system. For instance, when a Hopf bifurcation occurs, after the bifurcation, they can only assess the leading eigenvalue of the limit cycle which results from the bifurcation. What would happen in a large network where a node has a high degree and for which it is known that the dynamics becomes more or less a noisy limit cycle? In that case, most likely, I suspect that the eigenvalue would be close to zero although no particular bifurcation is involved. The kind of bifurcation which can be tracked with this technique seems to be quite limited to strong bifurcation like those destabilizing a singular point. In fact, the very basic introduction to

bifurcations inserted in this manuscript is limited to those observed in singular points. Would it be possible to extend the present technique to bifurcation on periodic orbit (the most frequently encountered in dynamical systems)?

Another question arises. Typically, the authors will be able to predict the destabilization of a stable singular point, but not its stabilization. This seems to be supported by Figs 6 and 7. This should be discussed.

The present technique is here tested for two very particular (strong) bifurcations arising on singular points. Typically, when a network would be on a stable singular point, there is no dynamics at all! So, to sum up, it is shown how a small network motif at rest could be destabilized. There is no indication that this technique will be still working with larger network and for nontrivial dynamics (I only mean here oscillating network). My feeling is that the amount of data is very large for the little gain discussed in this manuscript. Until an evidence on larger network ($N = 100$ for instance), it is rather hazardous to claim for any application to real network. The bifurcation of periodic orbits should be also investigated.

Other remarks

“some form of long-term behavior (stationarity, periodicity etc.)”. Stationarity designates in the sense of Kolmogorov a behavior whose mean quantities are constant. Consequently, periodic or chaotic behaviors are stationary. Please clarify this sentence. In particular, what is here designated as a stationary state is in fact an equilibrium, a steady state, a fixed point or even a singular point in a more mathematical terminology.

I suppose that by “fold bifurcation” the authors mean saddle-node or tangent bifurcation. It should be clear that most of these bifurcations are also encountered on periodic orbits.

Typically, after a Hopf bifurcation (the common one), the oscillations are sustained since there is a limit cycle induced by such a bifurcation.

“[...] while others lead to a catastrophic (and potentially irreversible) departure from the steady state.” Please provide here an example. In which sense is it catastrophic?

In general, I found the short introduction on bifurcations rather inaccurate.

The network considered in this paper are not “multilayer network” but “dynamical network” in which there is one variable per node. These dynamical networks are of a relatively low dimension ($N \leq 15$). This is a very strong limitation to this work.

Fig. 2. Transcritical ? Transcritical

There is no evidence in this manuscript that the systems investigated have multiple timescales. How a system having a stable singular point for solution could have multiple timescales? Please explain.

“Particularly interesting in this respect is the observed increase in accuracy close to the bifurcation point.” Unclear. Please clarify.

Board member pre-assessment comments (if available):

A closed form analytical solution for calculation of Jacobian matrix from time series is presented for a system near its steady state. Since the Jacobian can give information about the stability of the system against perturbations its leading eigenvalue can be used for early warning signal for critical transitions. As such, the method has relevance in data analysis and will be of practical application once the data demand can be reduced. The method is applied to multilayer networks of an ecological meta-food web model as an illustration.

Author's Response to Decision Letter for (RSPA-2019-0767.R0)

See Appendix A.

RSPA-2020-0742.R0

Review form: Referee 1

Is the manuscript an original and important contribution to its field?

Good

Is the paper of sufficient general interest?

Good

Is the overall quality of the paper suitable?

Good

Can the paper be shortened without overall detriment to the main message?

Yes

Do you think some of the material would be more appropriate as an electronic appendix?

No

Do you have any ethical concerns with this paper?

No

Recommendation?

Accept with minor revision (please list in comments)

Comments to the Author(s)

The authors have done a fine job of revising this manuscript. I still feel quite strongly that they need to acknowledge potential biases arising from incorrectly assuming that interactions are zero. Non-consumptive interactions are particularly hard to identify without direct field observations as is the spatial scales over which populations are effectively coupled. This is especially true when observations are made over coarse discrete time steps, such as the annual sampling that is prevalent in ecological monitoring. Although it would be ideal for the authors to evaluate the effects of incorrectly placed zeros numerically, I would settle for a paragraph in the discussion expanding on possible problems arising when zeros are not certain and assigned incorrectly.

Review form: Referee 3

Is the manuscript an original and important contribution to its field?

Good

Is the paper of sufficient general interest?

Good

Is the overall quality of the paper suitable?

Good

Can the paper be shortened without overall detriment to the main message?

Yes

Do you think some of the material would be more appropriate as an electronic appendix?

No

Do you have any ethical concerns with this paper?

No

Recommendation?

Accept with minor revision (please list in comments)

Comments to the Author(s)

In summary, I enjoyed reading this paper and believe it will make an interesting contribution to the field. I thought it was well written and clear. Its practical use is not yet clear, but it does represent a mathematical development with much potential.

To be clear to the authors, I was asked to review only in this second round – I don't know if that is because I am a 'third opinion' or a replacement for one of the earlier reviewers.

As it has already gone through a round of revision, I believe it is essentially acceptable as it is (bar a few minor typos). However, I think there are a number of areas where the manuscript could be significantly developed to better contextualise the results. I should note that my comments come largely from the perspective of an ecologist rather than a physicist. I appreciate that this is Proc A, not Proc B, but there are several lines along which the biological significance of the results could be helpfully elaborated, if the authors wished to meaningfully connect with a wider audience. I will have to trust to the previous reviewers that the algebra is correct – while I could happily follow the chain of logic, I'm afraid meaningfully checking those proofs is beyond me.

1. As the previous reviewers and the authors note, the data requirements of the method as presented in this paper are enormous. To their credit the authors make a point of discussing this limitation, even in the abstract, but don't directly discuss how the quality of the estimates declines to shorter time series. An analysis presenting how estimation error is related to time series length would give at least some indication of its plausibility in real systems (although granted, it will be highly system dependent). A discussion of how data length relates to uncertainties in the estimation of the covariance matrix Γ , and through to the estimation of J could be helpful here. Understanding the confidence required in the covariance matrix could be useful staging post in this, rather than focussing purely on time series length.

2. Some explicit comparison to other approaches to estimate Jacobians from time series (e.g. Sugihara et al. Science (2012)) would be very helpful. This was suggested by the previous reviewer 1, but from what I can gather was only marginally acted upon. While I appreciate the author's response-to-reviewers point (the unique value of analytic reconstructions of the Jacobian for the purposes of downstream work), this did not seem to be the main thrust of the manuscript, and shouldn't preclude a discussion of existing similar work.

In a similar vein, since other authors have also attempted to extract measures of stability from time series (e.g. Ives et al (2003), Estimating community stability and ecological interactions from time-series data. Ecol. Mono., 73: 301-330; Cenci & Saavedra (2019). Non-parametric estimation of the structural stability of non-equilibrium community dynamics. Nat. Ecol. Evol. 3, 912-918) it would be informative to the reader to more explicitly compare and contrast the methods in the manuscript with pre-existing methods, particularly if the intended audience is physicists who may be unaware of the large body of ecological work on this topic. At present the review element of the manuscript focusses, perhaps overly exclusively, on other methods for early warning signals.

3. The application of their method to the metacommunity case did raise some questions with me. Imagine a case where there was relatively little migration between communities (perhaps just one species disperses a little between sites, such that the systems although

technically dynamically 'linked', are effectively independent). In this situation, my instinct would be that it would be easier to try and understand the dynamics of each local community independently. But the headline results here suggest the opposite - it could be better to consider the system as one large system, because then a larger fraction of the overall system could be prespecified as zero.

I guess my question is, is this really the case? And if not, is it possible to say anything about how tightly coupled systems have to be in order to make this - larger-is-easier phenomenon occur?

I appreciate that this particular question comes from the particular metacommunity case, but the pattern of modularity in dynamical networks that creates this challenge is quite common.

4. Finally, the authors declare that no data was used and do not present any of their code. While strictly speaking true, it could be of considerable benefit to future development of this method if the authors included their code, to allow future uses to compare and check their implementations.

Minors:

In terms of the back-and-forth between the reviewer 2 and the authors, I think it reasonable to describe the system as a multilayer network. There are two distinct types of population interaction - the predation and the dispersal. But as the authors say, it is largely semantics. Representing metacommunities in an expanded matrix format like this is not unusual (e.g. Gravel, Massol. & Leibold. Stability and complexity in model meta-ecosystems. *Nat Commun* 7, 12457 (2016)).

Pg 5 line 51 - It can also affect the dynamics of other species at the patch where it is currently at, e.g. by migration. - should be 'by competition'?

Pg 9 line 36 spelling of 'spatial'

Pg 13 line 51. 'to the an'

p. 13. Line 45. 'predator-dependence of the predation rate' could do with some more explaining.

Figure 3 caption does not explain what ϕ is or the significance of the shading.

Finally, the SI could benefit from a spell check

Decision letter (RSPA-2020-0742.R0)

02-Feb-2021

Dear Dr Gross,

On behalf of the Editor, I am pleased to inform you that your Manuscript RSPA-2020-0742 entitled "A closed form for Jacobian reconstruction from timeseries and its application as an early warning signal in network dynamics" has been accepted for publication subject to minor revisions in Proceedings A. Please find the referees' comments below.

The reviewer(s) have recommended publication, but also suggest some minor revisions to your manuscript. Therefore, I invite you to respond to the reviewer(s)' comments and revise your manuscript. Please note that we have a strict upper limit of 28 pages for each paper. Please endeavour to incorporate any revisions while keeping the paper within journal limits. Please note that page charges are made on all papers longer than 20 pages. If you cannot pay these charges you must reduce your paper to 20 pages before submitting your revision. Your paper has been ESTIMATED to be 17 pages. We cannot proceed with typesetting your paper without your agreement to meet page charges in full should the paper exceed 20 pages when typeset. If you have any questions, please do get in touch.

It is a condition of publication that you submit the revised version of your manuscript within 7 days. If you do not think you will be able to meet this date please let me know in advance of the due date.

To revise your manuscript, log into <https://mc.manuscriptcentral.com/prsa> and enter your Author Centre, where you will find your manuscript title listed under "Manuscripts with Decisions." Under "Actions," click on "Create a Revision." Your manuscript number has been appended to denote a revision.

You will be unable to make your revisions on the originally submitted version of the manuscript. Instead, revise your manuscript and upload a new version through your Author Centre.

When submitting your revised manuscript, you will be able to respond to the comments made by the referee(s) and upload a file "Response to Referees" in Step 1: "View and Respond to Decision Letter". You can use this to document any changes you make to the original manuscript. In order to expedite the processing of the revised manuscript, please be as specific as possible in your response to the referee(s).

IMPORTANT: Your original files are available to you when you upload your revised manuscript. Please delete any redundant files before completing the submission process.

In addition to addressing all of the reviewers' and editor's comments, your revised manuscript **MUST** contain the following sections before the reference list (for any heading that does not apply to your work, please include a comment to this effect):

- Acknowledgements
- Funding statement

See <https://royalsociety.org/journals/authors/author-guidelines/> for further details.

When uploading your revised files, please make sure that you include the following as we cannot proceed without these:

- 1) A text file of the manuscript (doc, txt, rtf or tex), including the references, tables (including captions) and figure captions. Please remove any tracked changes from the text before submission. PDF files are not an accepted format for the "Main Document".
- 2) A separate electronic file of each figure (tif, eps or print-quality pdf preferred). The format should be produced directly from original creation package, or original software format.
- 3) Electronic Supplementary Material (ESM): all supplementary materials accompanying an accepted article will be treated as in their final form. Note that the Royal Society will not edit or typeset supplementary material and it will be hosted as provided. Please ensure that the supplementary material includes the paper details where possible (authors, article title, journal name). Supplementary files will be published alongside the paper on the journal website and posted on the online figshare repository (<https://figshare.com>). The heading and legend provided for each supplementary file during the submission process will be used to create the figshare page, so please ensure these are accurate and informative so that your files can be found in searches. Files on figshare will be made available approximately one week before the accompanying article so that the supplementary material can be attributed a unique DOI.

Alternatively you may upload a zip folder containing all source files for your manuscript as described above with a PDF as your "Main Document". This should be the full paper as it appears when compiled from the individual files supplied in the zip folder.

Article Funder

Please ensure you fill in the Article Funder question on page 2 to ensure the correct data is collected for FundRef (<http://www.crossref.org/fundref/>).

Media summary

Please ensure you include a short non-technical summary (up to 100 words) of the key findings/importance of your paper. This will be used for to promote your work and marketing purposes (e.g. press releases). The summary should be prepared using the following guidelines:

- *Write simple English: this is intended for the general public. Please explain any essential technical terms in a short and simple manner.
- *Describe (a) the study (b) its key findings and (c) its implications.
- *State why this work is newsworthy, be concise and do not overstate (true 'breakthroughs' are a rarity).
- *Ensure that you include valid contact details for the lead author (institutional address, email address, telephone number).

Cover images

We welcome submissions of images for possible use on the cover of Proceedings A. Images should be square in dimension and please ensure that you obtain all relevant copyright permissions before submitting the image to us. If you would like to submit an image for consideration please send your image to proceedingsa@royalsociety.org

Once again, thank you for submitting your manuscript to Proceedings A and I look forward to receiving your revision. If you have any questions at all, please do not hesitate to get in touch.

Best wishes
 Raminder Shergill
proceedingsa@royalsociety.org
 Proceedings A

Reviewer(s)' Comments to Author:

Referee: 1

Comments to the Author(s)

The authors have done a fine job of revising this manuscript. I still feel quite strongly that they need to acknowledge potential biases arising from incorrectly assuming that interactions are zero. Non-consumptive interactions are particularly hard to identify without direct field observations as is the spatial scales over which populations are effectively coupled. This is especially true when observations are made over coarse discrete time steps, such as the annual sampling that is prevalent in ecological monitoring. Although it would be ideal for the authors to evaluate the effects of incorrectly placed zeros numerically, I would settle for a paragraph in the discussion expanding on possible problems arising when zeros are not certain and assigned incorrectly.

Referee: 3

Comments to the Author(s)

In summary, I enjoyed reading this paper and believe it will make an interesting contribution to the field. I thought it was well written and clear. Its practical use is not yet clear, but it does represent a mathematical development with much potential.

To be clear to the authors, I was asked to review only in this second round – I don't know if that is because I am a 'third opinion' or a replacement for one of the earlier reviewers.

As it has already gone through a round of revision, I believe it is essentially acceptable as it is (bar a few minor typos). However, I think there are a number of areas where the manuscript could be significantly developed to better contextualise the results. I should note that my comments come largely from the perspective of an ecologist rather than a physicist. I appreciate that this is Proc A, not Proc B, but there are several lines along which the biological significance of the results could

be helpfully elaborated, if the authors wished to meaningfully connect with a wider audience. I will have to trust to the previous reviewers that the algebra is correct – while I could happily follow the chain of logic, I’m afraid meaningfully checking those proofs is beyond me.

1. As the previous reviewers and the authors note, the data requirements of the method as presented in this paper are enormous. To their credit the authors make a point of discussing this limitation, even in the abstract, but don’t directly discuss how the quality of the estimates declines to shorter time series. An analysis presenting how estimation error is related to time series length would give at least some indication of its plausibility in real systems (although granted, it will be highly system dependent). A discussion of how data length relates to uncertainties in the estimation of the covariance matrix Γ , and through to the estimation of J could be helpful here. Understanding the confidence required in the covariance matrix could be useful staging post in this, rather than focussing purely on time series length.

2. Some explicit comparison to other approaches to estimate Jacobians from time series (e.g. Sugihara et al. *Science* (2012)) would be very helpful. This was suggested by the previous reviewer 1, but from what I can gather was only marginally acted upon. While I appreciate the author’s response-to-reviewers point (the unique value of analytic reconstructions of the Jacobian for the purposes of downstream work), this did not seem to be the main thrust of the manuscript, and shouldn’t preclude a discussion of existing similar work.

In a similar vein, since other authors have also attempted to extract measures of stability from time series (e.g. Ives et al (2003), *Estimating community stability and ecological interactions from time-series data. Ecol. Mono.*, 73: 301-330; Cenci & Saavedra (2019). *Non-parametric estimation of the structural stability of non-equilibrium community dynamics. Nat. Ecol. Evol.* 3, 912–918) it would be informative to the reader to more explicitly compare and contrast the methods in the manuscript with pre-existing methods, particularly if the intended audience is physicists who may be unaware of the large body of ecological work on this topic. At present the review element of the manuscript focusses, perhaps overly exclusively, on other methods for early warning signals.

3. The application of their method to the metacommunity case did raise some questions with me. Imagine a case where there was relatively little migration between communities (perhaps just one species disperses a little between sites, such that the systems although technically dynamically ‘linked’, are effectively independent). In this situation, my instinct would be that it would be easier to try and understand the dynamics of each local community independently. But the headline results here suggest the opposite – it could be better to consider the system as one large system, because then a larger fraction of the overall system could be prespecified as zero. I guess my question is, is this really the case? And if not, is it possible to say anything about how tightly coupled systems have to be in order to make this – larger-is-easier phenomenon occur? I appreciate that this particular question comes from the particular metacommunity case, but the pattern of modularity in dynamical networks that creates this challenge is quite common.

4. Finally, the authors declare that no data was used and do not present any of their code. While strictly speaking true, it could be of considerable benefit to future development of this method if the authors included their code, to allow future uses to compare and check their implementations.

Minors:

In terms of the back-and-forth between the reviewer 2 and the authors, I think it reasonable to describe the system as a multilayer network. There are two distinct types of population interaction - the predation and the dispersal. But as the authors say, it is largely semantics. Representing metacommunities in an expanded matrix format like this is not unusual (e.g. Gravel, Massol. & Leibold. *Stability and complexity in model meta-ecosystems. Nat Commun* 7, 12457 (2016)).

Pg 5 line 51 - It can also affect the dynamics of other species at the patch where it is currently at, e.g. by migration. – should be ‘by competition’?

Pg 9 line 36 spelling of ‘spatial’

Pg 13 line 51. ‘to the an’

p. 13. Line 45. ‘predator-dependence of the predation rate’ could do with some more explaining.

Figure 3 caption does not explain what ϕ is or the significance of the shading.

Finally, the SI could benefit from a spell check

Author's Response to Decision Letter for (RSPA-2020-0742.R0)

See Appendix B.

Decision letter (RSPA-2020-0742.R1)

18-Feb-2021

Dear Dr Gross

I am pleased to inform you that your manuscript entitled "A closed form for Jacobian reconstruction from timeseries and its application as an early warning signal in network dynamics" has been accepted in its final form for publication in Proceedings A.

Our Production Office will be in contact with you in due course. You can expect to receive a proof of your article soon. Please contact the office to let us know if you are likely to be away from e-mail in the near future. If you do not notify us and comments are not received within 5 days of sending the proof, we may publish the paper as it stands.

Open access

You are invited to opt for open access, our author pays publishing model. Payment of open access fees will enable your article to be made freely available via the Royal Society website as soon as it is ready for publication. For more information about open access please visit <https://royalsociety.org/journals/authors/which-journal/open-access/>. The open access fee for this journal is £1700/\$2380/€2040 per article. VAT will be charged where applicable.

Note that if you have opted for open access then payment will be required before the article is published – payment instructions will follow shortly.

If you wish to opt for open access then please inform the editorial office (proceedingsa@royalsociety.org) as soon as possible.

Your article has been estimated as being 17 pages long. Our Production Office will inform you of the exact length at the proof stage.

Proceedings A levies charges for articles which exceed 20 printed pages. (based upon approximately 540 words or 2 figures per page). Articles exceeding this limit will incur page charges of £150 per page or part page, plus VAT (where applicable).

Under the terms of our licence to publish you may post the author generated postprint (ie. your accepted version not the final typeset version) of your manuscript at any time and this can be made freely available. Postprints can be deposited on a personal or institutional website, or a recognised server/repository. Please note however, that the reporting of postprints is subject to a media embargo, and that the status the manuscript should be made clear. Upon publication of the definitive version on the publisher's site, full details and a link should be added.

You can cite the article in advance of publication using its DOI. The DOI will take the form: 10.1098/rspa.XXXX.YYYY, where XXXX and YYYY are the last 8 digits of your manuscript number (eg. if your manuscript number is RSPA-2017-1234 the DOI would be 10.1098/rspa.2017.1234).

For tips on promoting your accepted paper see our blog post:
<https://royalsociety.org/blog/2020/07/promoting-your-latest-paper-and-tracking-your-results/>

On behalf of the Editor of Proceedings A, we look forward to your continued contributions to the Journal.

Sincerely,
Raminder Shergill
proceedingsa@royalsociety.org

Appendix A

Response to Referees

Referee: 1

The authors present a novel method for estimating Jacobians from time series data. To my eye the main methodological novelty, apart from some clever linear algebra, is the use of structural zeros in the estimation.

Yes, this is exactly the core of the advance presented in the manuscript.

However, it seems to me that the authors are fairly casual in their assumption that sufficient structural zeros can be known with certainty a priori. This is particularly relevant to ecological data where a) behaviorally mediated indirect effects are difficult to identify a priori and b) data are often sampled over fairly coarse time steps. As such, I think the authors need to rigorously evaluate the sensitivity of their approach to incorrect placement of structural zeros.

We have restructured the explanations in the paper to show more clearly that this won't be a problem in practice. For typical use-cases there is no uncertainties involved as a sufficient number of zeroes can be inferred from hard information constraints. For illustration consider a tritrophic food chain. Suppose furthermore that we do not know the dispersal patterns precisely. However, it seems safe to assume that a prey individual does not have information about the predator density at a patch where the individual is not at. Hence we can assume that the population dynamics of prey is independent of the predator density in all other patches. In an P -patch system this gives us $2P(P-1)$ zeros. Applying the same reasoning to the other two species yields the same number of zeroes. Thus the number of known zeroes is now $6P(P-1)$. The total number of zeroes that need to be know in a three species system with P patches is $3P(3P-1)/2$. Comparing the two numbers one finds that there is a sufficient number of zeroes for any system that has at least 3 patches. This is a very harmless constraint and for larger food webs the condition become even simpler to justify.

The discussion of this point has been considerably expanded and moved from the conclusions to the section where the zeroes are first mentioned.

Secondarily, there are several extant approaches to inferring Jacobians from time series data (e.g. dynamic linear models, autoregressive models, etc). It seems relevant to ask how the new method performs compared to these alternatives.

This is indeed an important point and we spent a significant amount of work exploring alternative approaches. While different approaches come from very different starting points they all end up with very similar equation systems to solve. In all cases the central difficulty is that at some point causality needs to be constructed from correlations. This step always requires the integration of some external knowledge such as the structural zeroes used in our case. To our knowledge all previous methods integrate this additional information using numerical methods, such as Kalman filters, optimization etc.

Thus, the selling point of the proposed approach is not that it is faster (depending on details of the implementation it could be) or that it is more accurate (given enough computational time optimization can produce fantastic results) but rather that it provides a closed form solution. In this way the proposed approach is qualitatively different from all previous work. By providing a closed form solution it lays the

foundation for further mathematical work and allows Jacobian reconstruction to be integrated in larger calculations.

We emphasize this point more strongly in the revised version of the manuscript.

Referee: 2

The aim of this manuscript is to propose a mixed technique half-based on some prior knowledge on the governing equations and half-based on data for predicting sudden, qualitative and potentially irreversible transitions which may occur in complex networks. When the network is sufficiently sparse, and knowing the topology of the network, the authors assess the non-zero elements of the Jacobian matrix from time series correlations to compute the leading eigenvalue of the so-obtained jacobian matrix to detect bifurcation of singular points. As expected with such an approach, the amount of data required for accurate results is at present still prohibitively high for any practical application to a network from the real world. Indeed, typically, this technique requires to compute correlation between time series of all related variables. Since each node is at least once linked with another one, this means that at least one variable is measured in each node. Moreover, to accurately capture correctly the dynamics, it should be also necessary to investigate the node dynamics, meaning that even more variables should be measured. This is never clearly stated, but the present techniques applied to a N -dimensional system (here N means the number of variables spanning the state space of the full network) requires to measure the N variables which is a huge constraint for investigating a network. This is then combined with the knowledge of the adjacency matrix of the network.

We agree with the referee's assessment of the advantages and disadvantages of the method.

What the authors are able to track is the occurrence of a bifurcation by detecting when the real part of the leading eigenvalue is close to zero. It is therefore erroneous to claim that there is a reconstruction of the jacobian matrix. This is only a technique for assessing the leading eigenvalue. There is no evidence in this paper that the full jacobian matrix is actually correctly estimated.

We are sorry for this oversight. We focussed on the leading eigenvalue merely because of its importance for bifurcations. We expected that all eigenvalues would be inferred with similar accuracy and in response to the reviewer we have now verified this. As a result we have added a new figure to the manuscript to illustrate the inference of non-leading eigenvalues and eigenvectors. We can thus state with confidence that the method reconstructs the full Jacobian matrix.

Moreover, as the authors mentioned themselves, their technique can only assess the leading eigenvalue associated with an “attractor”, that is, with the trajectory actually followed by the system. For instance, when a Hopf bifurcation occurs, after the bifurcation, they can only assess the leading eigenvalue of the limit cycle which results from the bifurcation.

Yes, the purpose of the method is to infer the Jacobian from data. Hence the reason why it cannot be applied to unstable states is not that the method wouldn't work there (it would) but that we cannot gather data on a state at which the system does not remain.

What would happen in a large network where a node has a high degree and for which it is known that the dynamics becomes more or less a noisy limit cycle? In that case, most likely, I suspect that the eigenvalue would be close to zero although no particular bifurcation is involved. The kind of bifurcation which can be tracked with this technique seems to be quite limited to strong bifurcation like those destabilizing a singular point.

We agree with this assessment. The detection of destabilization of singular points is exactly the problem that we address. We would like to point out that this is a very important problem in a wide range of applications. Nevertheless, the Jacobian matrices reconstructed by the method contain more information than just stability and could also be used in other ways, e.g. for optimal control of engineered systems.

In fact, the very basic introduction to bifurcations inserted in this manuscript is limited to those observed in singular points. Would it be possible to extend the present technique to bifurcation on periodic orbit (the most frequently encountered in dynamical systems)?

Yes, and we think this extension is relatively straight forward. For example the method could be directly applied to Poincare maps or stroboscopic projections of a periodic system. We have added a discussion to the conclusions of the manuscript.

Another question arises. Typically, the authors will be able to predict the destabilization of a stable singular point, but not its stabilization. This seems to be supported by Figs 6 and 7. This should be discussed.

Yes, again this depends on data availability. If the system resides in the vicinity of the stabilized state we should see the stabilization. If the state of the system is far away we wouldn't see it. Some comments on this matter have been added.

The present technique is here tested for two very particular (strong) bifurcations arising on singular points. Typically, when a network would be on a stable singular point, there is no dynamics at all! So, to sum up, it is shown how a small network motif at rest could be destabilized. There is no indication that this technique will be still working with larger network and for nontrivial dynamics (I only mean here oscillating network). My feeling is that the amount of data is very large for the little gain discussed in this manuscript. Until an evidence on larger network ($N = 100$ for instance), it is rather hazardous to claim for any application to real network. The bifurcation of periodic orbits should be also investigated.

We believe this comment may be rooted in a misunderstanding of the model: We already provide a 60-dimensional example.

Moreover, our mathematical reasoning holds independently of network size. Accuracy should actually be better for a larger system. We would also point out that for SDE-based dynamical networks the systems considered in the paper are actually quite

large. What limits the examples in the paper is not the actual reconstruction but the data generation. Simulating 60 coupled nonlinear SDEs with significant timescale separation is still a sizable numerical task.

Other remarks

“some form of long-term behavior (stationarity, periodicity etc.)”. Stationarity designates in the sense of Kolmogorov a behavior whose mean quantities are constant. Consequently, periodic or chaotic behaviors are stationary. Please clarify this sentence. In particular, what is here designated as a stationary state is in fact an equilibrium, a steady state, a fixed point or even a singular point in a more mathematical terminology.

This has been clarified

I suppose that by “fold bifurcation” the authors mean saddle-node or tangent bifurcation. It should be clear that most of these bifurcations are also encountered on periodic orbits.

We now mention the alternative names for this bifurcation

Typically, after a Hopf bifurcation (the common one), the oscillations are sustained since there is a limit cycle induced by such a bifurcation.

Why this is true for some common small models. The supercritical Hopf bifurcation is exactly as common as the subcritical form. So, we cannot claim that there will generally be sustained oscillations.

“[...] while others lead to a catastrophic (and potentially irreversible) departure from the steady state.” Please provide here an example. In which sense is it catastrophic?

Catastrophic in the sense of catastrophe theory, meaning discontinuous, we have clarified this.

In general, I found the short introduction on bifurcations rather inaccurate.

This introduction is intended for readers not familiar with the subject and thus avoids mathematical jargon. Nevertheless we have confirmed that the information provided is accurate and not misleading.

The network considered in this paper are not “multilayer network” but “dynamical network” in which there is one variable per node. These dynamical networks are of a relatively low dimension ($N < 15$). This is a very strong limitation to this work.

This is semantics, but we have to disagree in this respect. There are 15 geographical nodes in the network, which does not mean that the dynamical dimension is 15 rather it is 15 times the number of species placed in the nodes. So in case of the 4-species web the system is 60 dimensional.

Regarding the reviewers' claim that our system is not a multilayer network, we would like to point out that the system is described as a multiplex. i.e. the different species correspond to different layers on top of the geographical network. Our previous work that uses the same model is highlighted as an important example of real world multilayer network in Prof. Bianconi's excellent book “Multilayer Networks: Structure and Function”

Fig. 2. Trascritical → Transcritical

Fixed

There is no evidence in this manuscript that the systems investigated have multiple timescales. How a system having a stable singular point for solution could have multiple timescales? Please explain.

We are referring to the theory of multiple timescale dynamical systems. A good current reference is Christian Kühn's recent book. What is meant is the time scale of the intrinsic dynamics of the various species. While these dynamics are not visible in a stationary state they become evident in the response after perturbation. In an ecological system the top predators have lifespans that span tens of years while primary producers life takes place on the scale of days or weeks. Correspondingly the upper layers of the food web respond several orders of magnitude slower to perturbations than the lower layers. This so-called allometric scaling is realistically modelled in our food webs. Which greatly increases the numerical effort as it leads to stiff systems.

"Particularly interesting in this respect is the observed increase in accuracy close to the bifurcation point." Unclear. Please clarify.

A reference to Fig. 7 from which this is one of the major conclusions has been added.

Board member pre-assessment comments (if available):

A closed form analytical solution for calculation of Jacobian matrix from time series is presented for a system near its steady state. Since the Jacobian can give information about the stability of the system against perturbations its leading eigenvalue can be used for early warning signal for critical transitions. As such, the method has relevance in data analysis and will be of practical application once the data demand can be reduced. The method is applied to multilayer networks of an ecological meta-food web model as an illustration.

We thank the board member for this encouraging assessment.

Appendix B

Dear Prof. Shergill,

Thank you very much for the positive decision on our manuscript. Please find a new version of the manuscript enclosed, in which we have implemented the reviewers recommendations.

Let me take this opportunity on behalf of all coauthors to thank you and the reviewers for the valuable and insightful reviewing process.

For the record I declare that I am happy to cover page charges if the final article should be longer than 20 pages.

Best regards
Thilo Gross

RESPONSE TO REVIEWERS

Referee: 1

Comments to the Author(s)

The authors have done a fine job of revising this manuscript. I still feel quite strongly that they need to acknowledge potential biases arising from incorrectly assuming that interactions are zero. Non-consumptive interactions are particularly hard to identify without direct field observations as is the spatial scales over which populations are effectively coupled. This is especially true when observations are made over coarse discrete time steps, such as the annual sampling that is prevalent in ecological monitoring. Although it would be ideal for the authors to evaluate the effects of incorrectly placed zeros numerically, I would settle for a paragraph in the discussion expanding on possible problems arising when zeros are not certain and assigned incorrectly.

We have added a new paragraph to the discussion (paragraph 2) that highlights this point.

Referee: 3

Comments to the Author(s)

In summary, I enjoyed reading this paper and believe it will make an interesting contribution to the field. I thought it was well written and clear. Its practical use is not yet clear, but it does represent a mathematical development with much potential.

To be clear to the authors, I was asked to review only in this second round – I don't know if that is because I am a 'third opinion' or a replacement for one of the earlier reviewers.

As it has already gone through a round of revision, I believe it is essentially acceptable as it is (bar a few minor typos). However, I think there are a number of areas where the manuscript

could be significantly developed to better contextualise the results. I should note that my comments come largely from the perspective of an ecologist rather than a physicist. I appreciate that this is Proc A, not Proc B, but there are several lines along which the biological significance of the results could be helpfully elaborated, if the authors wished to meaningfully connect with a wider audience. I will have to trust to the previous reviewers that the algebra is correct – while I could happily follow the chain of logic, I'm afraid meaningfully checking those proofs is beyond me.

We thank the reviewer for the thorough reading of the manuscript and the insightful report.

1. As the previous reviewers and the authors note, the data requirements of the method as presented in this paper are enormous. To their credit the authors make a point of discussing this limitation, even in the abstract, but don't directly discuss how the quality of the estimates declines to shorter time series. An analysis presenting how estimation error is related to time series length would give at least some indication of its plausibility in real systems (although granted, it will be highly system dependent). A discussion of how data length relates to uncertainties in the estimation of the covariance matrix Γ , and through to the estimation of J could be helpful here. Understanding the confidence required in the covariance matrix could be useful staging post in this, rather than focussing purely on time series length.

There is a simple answer to this question and a much longer one. The simple is that errors in covariance should propagate almost linearly to errors in the Jacobian. We now mention this in the second paragraph of the discussion. But we fully agree with the reviewer that the impact of the true Jacobian on error propagation merits a thorough investigation. One advantage of the formula derived in the present manuscript is that this propagation can now be explored mathematically with matrix perturbation theory. We have therefore chosen not to present a feel-good numerical bootstrap analysis of this error as we expect this would be more hindrance than help to deeper analysis in the future.

2. Some explicit comparison to other approaches to estimate Jacobians from time series (e.g. Sugihara et al. Science (2012)) would be very helpful. This was suggested by the previous reviewer 1, but from what I can gather was only marginally acted upon. While I appreciate the author's response-to-reviewers point (the unique value of analytic reconstructions of the Jacobian for the purposes of downstream work), this did not seem to be the main thrust of the manuscript, and shouldn't preclude a discussion of existing similar work.

In a similar vein, since other authors have also attempted to extract measures of stability from time series (e.g. Ives et al (2003), Estimating community stability and ecological interactions from time-series data. Ecol. Mono., 73: 301-330; Cenci & Saavedra (2019). Non-parametric estimation of the structural stability of non-equilibrium community dynamics. Nat. Ecol. Evol. 3, 912–918) it would be informative to the reader to more explicitly compare and contrast the methods in the manuscript with pre-existing methods, particularly if the intended audience is physicists who may be unaware of the large body of ecological work on this topic. At present the

review element of the manuscript focusses, perhaps overly exclusively, on other methods for early warning signals.

We thank the reviewer for pointing out these citations. We cite them in the revise paper and discuss Sugihara et al. in more detail as we feel it is the most relevant.

3. The application of their method to the metacommunity case did raise some questions with me. Imagine a case where there was relatively little migration between communities (perhaps just one species disperses a little between sites, such that the systems although technically dynamically 'linked', are effectively independent). In this situation, my instinct would be that it would be easier to try and understand the dynamics of each local community independently. But the headline results here suggest the opposite – it could be better to consider the system as one large system, because then a larger fraction of the overall system could be prespecified as zero. I guess my question is, is this really the case? And if not, is it possible to say anything about how tightly coupled systems have to be in order to make this – larger-is-easier phenomenon occur?

I appreciate that this particular question comes from the particular metacommunity case, but the pattern of modularity in dynamical networks that creates this challenge is quite common.

Using additional information, such as the observation that local communities are identical would make the task easier. If such additional information is not used, applying the reconstruction to the whole system instead of local communities does indeed improve the quality of reconstruction although it makes the calculation numerically more demanding (though this is hardly an obstacle in this case). In the limit where the system is disconnected the additional 0s one gains from considering the whole system don't add value as they contribute only trivial conditions.

4. Finally, the authors declare that no data was used and do not present any of their code. While strictly speaking true, it could be of considerable benefit to future development of this method if the authors included their code, to allow future uses to compare and check their implementations.

We have added a reference to a code repository, where the codes will be uploaded.

Minors:

In terms of the back-and-forth between the reviewer 2 and the authors, I think it reasonable to describe the system as a multilayer network. There are two distinct types of population interaction - the predation and the dispersal. But as the authors say, it is largely semantics. Representing metacommunities in an expanded matrix format like this is not unusual (e.g. Gravel, Massol. & Leibold. Stability and complexity in model meta-ecosystems. Nat Commun 7, 12457 (2016)).

Pg 5 line 51 - It can also affect the dynamics of other species at the patch where it is currently at, e.g. by migration. – should be 'by competition'?

Done. Thanks for spotting this.

Pg 9 line 36 spelling of 'spatial'

Fixed

Pg 13 line 51. 'to the an'

Fixed

p. 13. Line 45. 'predator-dependence of the predation rate' could do with some more explaining.

This has been rephrased to provide more detail

Figure 3 caption does not explain what ϕ is or the significance of the shading.

Fixed

Finally, the SI could benefit from a spell check

We have carefully checked the SI